# The effect of environmental information on evolution of cooperation in stochastic games

**Maria Kleshnina** [1,7] ✉, **Christian Hilbe** [2,7], **Štěpán Šimsa** [3,4], **Krishnendu Chatterjee**[3] **& Martin A. Nowak** [5,6]

Many human interactions feature the characteristics of social dilemmas where individual actions have consequences for the group and the environment. The feedback between behavior and environment can be studied with the framework of stochastic games. In stochastic games, the state of the environment can change, depending on the choices made by group members. Past work suggests that such feedback can reinforce cooperative behaviors. In particular, cooperation can evolve in stochastic games even if it is infeasible in each separate repeated game. In stochastic games, participants have an interest in conditioning their strategies on the state of the environment. Yet in many applications, precise information about the state could be scarce. Here, we study how the availability of information (or lack thereof) shapes evolution of cooperation. Already for simple examples of two state games we find surprising effects. In some cases, cooperation is only possible if there is precise information about the state of the environment. In other cases, cooperation is most abundant when there is no information about the state of the environment. We systematically analyze all stochastic games of a given complexity class, to determine when receiving information about the environment is better, neutral, or worse for evolution of cooperation.

Cooperation can be conceptualized as an individually costly behavior that creates a benefit to others[1]. Such cooperative behaviors have evolved in many species, from uni-cellular organisms to mammals[2]. Yet they are arguably most abundant and complex in humans, where they form the very basis of families, institutions, and society[3,4]. Humans often support cooperation through direct reciprocity[5]. Here, people preferentially help those who have been helpful in the past[6]. Such forms of direct reciprocity naturally emerge when groups are stable, and when cooperation yields substantial returns[7]. In that case, individuals readily learn to engage in conditional cooperation, using strategies like Tit-for-tat[8–11] (TFT), Win-Stay Lose-Shift[12,13] (WSLS), or multiplayer variants thereof[14–16]. When everyone adopts these strategies, groups can sustain cooperation despite any short-run incentives to free ride[17,18].

To describe direct reciprocity formally, traditional models of cooperation consider individuals who face the same strategic interaction (game) over and over again. The most prominent model of this kind is the iterated prisoner's dilemma[8]. In this game, two individuals (players) repeatedly decide whether to cooperate or defect. While the players' decisions may change from one round to the next, the feasible payoffs remain constant. Models based on iterated games have become fundamental for our understanding of reciprocity. However, they presume that interactions take place in a constant social and natural environment. Individual actions in one round have no effect on the exact game being played in future. In contrast, in many applications, the environment is adaptive, such as when populations aim to control an epidemics[19–21], manage natural resources[22–24], or mitigate climate change[25–27]. Changing environments in turn often bring about a

[1]Institute for Advanced Study in Toulouse, Toulouse, France. [2]Max Planck Research Group Dynamics of Social Behavior, Max Planck Institute for Evolutionary Biology, Plön, Germany. [3]IST Austria, Klosterneuburg, Austria. [4]Faculty of Mathematics and Physics, Charles University, Prague, Czech Republic. [5]Department of Mathematics, Harvard University, Cambridge, MA, USA. [6]Department of Organismic and Evolutionary Biology, Harvard University, Cambridge, MA, USA. [7]These authors contributed equally: Maria Kleshnina, Christian Hilbe. ✉e-mail: maria.kleshnina@iast.fr

change in the exact game being played. Such applications are therefore best described with models in which there is a feedback between behavior and environment. In the context of direct reciprocity, such feedbacks can be incorporated with the framework of stochastic games[28–30].

In stochastic games, individuals interact over multiple time periods. Each period, the players' environment is in one of several possible states. This state can change from one period to the next, depending on the current state, the players' actions, and on chance. Changes of the state affect the players' available strategies and their feasible payoffs. In this way, stochastic games are better able to describe social dilemmas in which individual actions affect the nature of a group's future interactions. Yet previous evolutionary models of stochastic games presume that individuals are perfectly aware of the current state[31–33]. This allows individuals to coordinate on appropriate responses once the state has changed. In contrast, in many applications, any knowledge about the state of the environment is at best incomplete. Such uncertainties can in turn have dramatic effects on human behavior[34–37]. Understanding the impact of information on decision-making has been a rich field of study in economics. Corresponding studies suggest that the effect of information is often positive, even though there are situations in which it has adverse effects[38–40]. Additionally, studies of partially observable stochastic games suggest that settings with incomplete information can benefit decision-makers[41,42].

In the following, we explore how state uncertainty in stochastic games shapes the evolution of cooperation. To this end, we compare two scenarios. First, we consider the case when individuals are able to learn the state of their environment and condition their decisions on the current state. We will refer to this case as the 'full-information setting'. In the second case, individuals may be aware that they are engaged in a stochastic game but they either ignore or are unable to obtain information about the current state. As a result, their decisions are independent of their environment. We refer to this case as the 'no-information setting'. To compare these two settings we focus on the simplest possible case, where two players may experience two possible states. Already for this elementary setup, we obtain an extremely rich family of models that gives rise to many different possible dynamics.

Already here, we observe that conditioning strategies on state information can have drastic effects on how people cooperate.

To quantify the importance of state information, we introduce a measure to which we refer as the 'value of information'. This value reflects by how much the cooperation rate in a population changes by gaining access to information about the present state. When this value is positive, access to information makes the population more cooperative. In that case, we speak of a 'benefit of information'. In general, it is also possible to observe negative values, in which case we speak of a 'benefit of ignorance'. With analytical methods for the important limit of weak selection[43–45], and with numerical computations for arbitrary selection strengths, we compare the value of information across many stochastic games. We identify settings where receiving information is better, neutral, or worse for the evolution of cooperation. Most often, information is highly beneficial. However, there are also a few notable exceptions in which populations can achieve more cooperation when they are ignorant of their state. In the following, we describe and characterize these cases in detail.

## Results

### Stochastic games with and without state information

To explore the dynamics of cooperation in variable environments, we consider stochastic games[31–33]. We introduce our framework for the most simple setup, in which the game takes place among two players who interact for infinitely many rounds, without discounting of their future payoffs. In each round, players can find themselves in two possible states, $S = \{s_1, s_2\}$. Depending on the state, players engage in one of two possible prisoner's dilemma games. In either game, they can either cooperate ($C$) or defect ($D$). Cooperation means to pay a cost $c$ for the other player to get a benefit $b_i$. The cost of cooperation is fixed, but the benefit $b_i$ depends on the present state $s_i$ (Fig. 1a). Without loss of generality, we assume that the first state is more profitable, such that $b_1 \geq b_2 > c := 1$. However, states can change from one round to the next, depending on the game's transition vector

$$\mathbf{q} = (q_{CC}^1, q_{CD}^1, q_{DD}^1; q_{CC}^2, q_{CD}^2, q_{DD}^2). \tag{1}$$

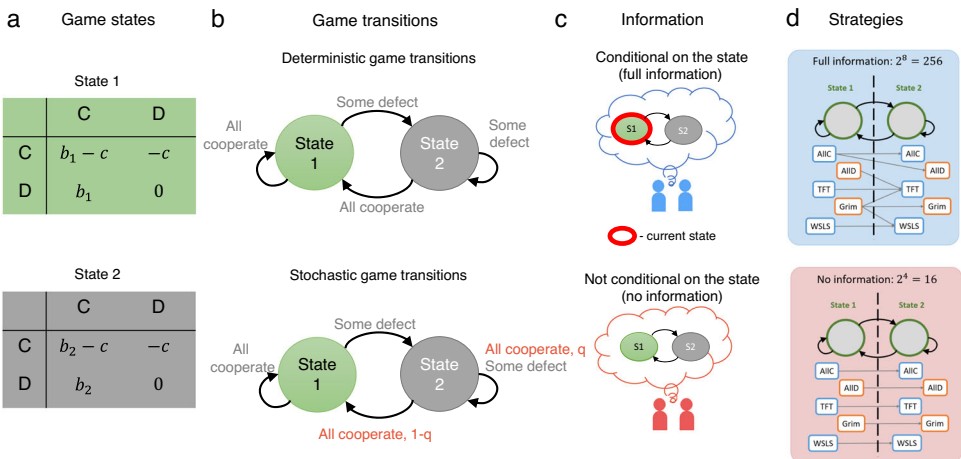

**Fig. 1 | Stochastic games with full and no information. a** We study 2-state stochastic games where transitions between the states depend on the players' actions. In each state, players engage in a prisoners' dilemma with benefit $b_1$ (or $b_2$) and cost $c$. The two benefit parameters $b_1$ and $b_2$ might reflect the group's environmental conditions. Without loss of generality, we assume $b_1 \geq b_2$. **b** Transitions between the states can be either completely determined by the player's current actions (deterministic game transitions), or they may additionally depend on chance events (stochastic game transitions). In the two cases depicted here, environmental conditions worsen when players defect, reducing the players' possible benefits. Once players resume mutual cooperation, they may return to the more profitable first state. We note that in the bottom case, only a single transition depends on chance; in that case, we speak of a single-stochastic transition vector. **c** In this work, we compare two possible scenarios, depending on whether or not players are able to condition their behavior on the current state ('full information' versus 'no information'). **d** With full information, individuals can react differently to their opponent, depending on the current state. As a result, they can choose among $2^8 = 256$ deterministic memory-one strategies. Without information, players need to act in the same way in each of the two states. Hence there are only $2^4 = 16$ deterministic memory-one strategies. The acronyms ALLC, ALLD, TFT, WSLS refer to unconditional cooperation, unconditional defection, tit-for-tat, and win-stay lose-shift, respectively.

Here, each entry $q_{a\tilde{a}}^i \in [0,1]$ is the probability that players find themselves in the more profitable state $s_1$ in the next round. This probability depends on the previous state $s_i$ and on the players' previous actions $a$ and $\tilde{a}$. For example, the transition vector $\mathbf{q} = (1, 0, 0; 1, 0, 0)$ corresponds to a game in which players are only in the more profitable state if they both cooperated in the previous round. Note that we assume the transition vector to be symmetric. That is, transition probabilities depend on the number of cooperators, but they are independent of who cooperated ($q_{CD}^i = q_{DC}^i$ for all $i$). We say a transition vector is deterministic if each entry $q_{a\tilde{a}}^i$ is either zero or one (Fig. 1b). Even for deterministic vectors we speak of a 'stochastic game', because games with deterministic transitions represent a special case of our framework. Based on Eq. (1), there are $2^6 = 64$ deterministic transition vectors in total. We call a transition vector single-stochastic if there is exactly one entry that is strictly between zero and one. Games with single-stochastic transitions can serve as the most elementary example of an interaction for which the environment depends on chance events.

To explore how often players cooperate depending on the information they have, we compare two settings (Fig. 1c). In the full-information setting, players learn the present state before making decisions. Thus, their strategies may depend on both the present state and on the players' actions in the previous rounds. Herein, we assume that players make decisions based on memory-1 strategies. Such strategies only take into account the outcome of the last round[46] (extensions to more complex strategies[47–52] are possible, but for simplicity we do not explore them here). In the full information setting, memory-1 strategies take the form of an 8-tuple,

$$\mathbf{p}_F = (p_{CC}^1, p_{CD}^1, p_{DC}^1, p_{DD}^1; p_{CC}^2, p_{CD}^2, p_{DC}^2, p_{DD}^2). \qquad (2)$$

Here, $p_{a\tilde{a}}^i$ is the player's probability to cooperate in state $s_i$, depending on the focal player's and the co-player's previous actions $a$ and $\tilde{a}$, respectively. We compare this full-information setting with a no-information setting, in which individuals are unable to condition their behavior on the current state. In that case, strategies are 4-tuples

$$\mathbf{p}_N = (p_{CC}, p_{CD}, p_{DC}, p_{DD}). \qquad (3)$$

We note that the set of no-information strategies is a strict subset of the full-information strategies (they correspond to those $\mathbf{p}_F$ for which $p_{a\tilde{a}}^1 = p_{a\tilde{a}}^2$ for all actions $a$ and $\tilde{a}$). For simplicity, we assume in the following that the players' strategies are deterministic, such that each entry is either zero or one. For full information, there are $2^8 = 256$ deterministic strategies. For no information, there are $2^4 = 16$ deterministic strategies. Some results for stochastic strategies are shown in Fig. S1a, b.

The players' strategies may be subject to errors with some small probability $\varepsilon$. This model parameter reflects the assumption that people may occasionally make mistakes when engaging in reciprocity[53,54]. In that case, an intended cooperation may be mis-implemented as a defection (and vice versa). Games with errors have the useful technical property that the long-run dynamics is independent of the players' initial moves[46]. For $\varepsilon > 0$, a player with strategy $\mathbf{p}$ effectively implements the strategy $(1 - \varepsilon)\mathbf{p} + \varepsilon(\mathbf{1} - \mathbf{p})$. In particular, even when the original strategy $\mathbf{p}$ is deterministic, the effective strategy is stochastic. Given the error probability, the players' strategies, and the game's transition vector, we can compute how often players cooperate on average and which payoffs they get (see Methods).

Because we are interested in how cooperation evolves, we do not consider players with fixed strategies. Rather players can change their strategies in time, depending on the payoffs they yield. To describe this evolutionary dynamics, we use a pairwise comparison process[55]. This process considers populations of fixed size $N$. Players receive payoffs by interacting with all other population members. At regular time intervals, one player is randomly chosen and given the opportunity to revise its strategy. The player may do so in two ways. With probability $\mu$, the player switches to a random deterministic memory-1 strategy (similar to a mutation in biological models of evolution). Otherwise, with probability $1 - \mu$, the focal player compares its own payoff $\pi$ to the payoff $\tilde{\pi}$ of a random role model. The player switches to the role model's strategy with probability $(1 + \exp[-\beta(\tilde{\pi} - \pi)])^{-1}$. The parameter $\beta > 0$ is the strength of selection. The higher this parameter, the more individuals are prone to imitate only those role models with a high payoff. Overall, these assumptions define a stochastic process on the space of all possible population compositions. For finite $\beta$, evolutionary trajectories do not converge to any particular outcome because no population composition is absorbing. However, because the process is ergodic, the respective time averages converge to an invariant distribution. This invariant distribution describes how often the population has a given composition in the long run (see Methods).

We study this evolutionary process analytically when mutations are rare and selection is weak (that is, when $\mu, \beta \to 0$). In addition, we numerically explore the process for arbitrary selection strengths. In either case, we compute which payoffs players receive on average and how likely they are to cooperate over time. By comparing the cooperation rates $\hat{\gamma}^F$ and $\hat{\gamma}^N$ for populations with full and no information, respectively, we quantify how favorable information is for the evolution of cooperation. We refer to the difference, $V_\beta(\mathbf{q}) := \hat{\gamma}^F - \hat{\gamma}^N$ as the value of (state) information. In general, this value depends on the game's transition vector $\mathbf{q}$, as well as on the strength of selection $\beta$. When this value is positive, populations achieve more cooperation when they learn the present state of the stochastic game.

In the following, we describe the results of this baseline model in detail. In the SI, we provide further results on the impact of different game parameters (Fig. S1), other strategy spaces (Fig. S2), and alternative learning rules (Fig. S3).

## The effect of state information in two examples

To begin with, we illustrate the effect of state information by exploring the dynamics of two examples. Both examples are variants of models that have been previously used to highlight the importance of stochastic games for the evolution of cooperation[31]. In the first example (Fig. 2a), players only remain in the more profitable first state if they both cooperate. If either of them defects, they transition to the inferior second state. Once there, they transition back to the more profitable state after one round, irrespective of the players' actions. The second state may thus be interpreted as a 'time-out'[31]. For numerical results, we assume that cooperation yields an intermediate benefit in the more profitable state and a low benefit in the inferior state ($b_1 = 1.8$, $b_2 = 1.3$).

When we simulate the evolutionary dynamics of this stochastic game, we observe that individuals consistently learn to cooperate when they have full information. In contrast, without information, they mostly defect (Fig. 2b). To explain this result, we numerically compute which strategies are most likely to evolve according to the process's invariant distribution, for each of the two cases (Fig. 2c). In the full-information setting, individuals predominantly adopt a strategy $\mathbf{p}_F = (1, 0, 0, 0; x, 0, 0, 1)$, where $x \in \{0, 1\}$ is arbitrary. This strategy may be considered as a variant of the WSLS rule that has been successful in the traditional prisoner's dilemma[12]. In particular, it is fully cooperative with itself. We prove in Supplementary Note 3 that this strategy forms a subgame perfect (Nash) equilibrium if $2b_1 - b_2 \geq 2c$, which is satisfied for the parameters we use (see also Fig. 3a). On the other hand, in the no-information setting, this strategy is no longer available. Instead, players can only sustain cooperation with the traditional WSLS rule $\mathbf{p}_N = (1, 0, 0, 1)$. This strategy is only an equilibrium under the more stringent condition $b_1 > 2c$. Because our parameters do not satisfy this

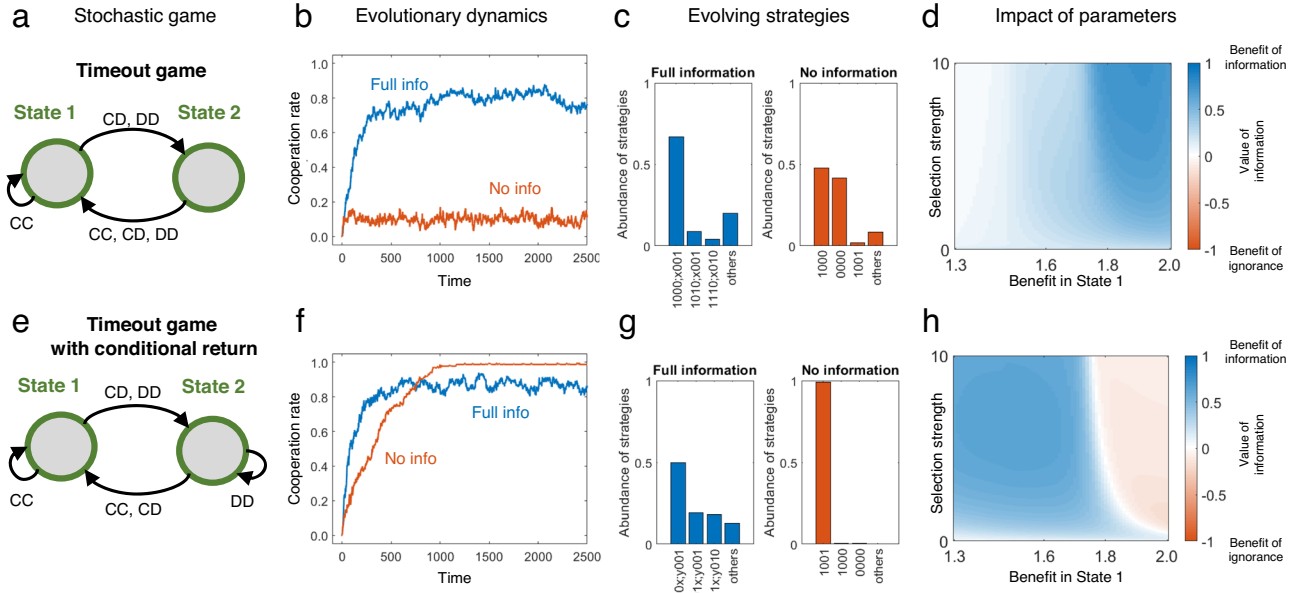

**Fig. 2 | A comparison of the value of information in two games. a**, **e** As an example, we consider the dynamics of two games with deterministic transitions. We refer to the first game with transition vector $\mathbf{q}_1 = (1, 0, 0; 1, 1, 1)$ as a game with timeout. The second game with transition vector $\mathbf{q}_2 = (1, 0, 0; 1, 1, 0)$ corresponds to a timeout game with conditional return. **b**, **f** To illustrate the evolutionary dynamics, we simulate the pairwise comparison process for both settings (full and no information). Full information yields more cooperation in the first game but less cooperation in the second. **c**, **g** To explore the impact of information in each game, we numerically compute the abundance of all strategies, according to the invariant distribution of the process (see Methods). In Fig. 3, we describe these abundances in more detail. **d**, **h** By simultaneously varying the benefit $b_1$ in the more profitable state and the selection strength $\beta$, we explore for which parameters there is a benefit of ignorance. Colors represent the value of information $V_\beta(\mathbf{q})$ according to the invariant distribution of the process. Default parameters: $b_1 = 1.8$, $b_2 = 1.3$, $c = 1$, population size $N = 100$, error rate $\varepsilon = 0.01$, and selection strength $\beta = 10$.

condition, cooperation does not evolve in the no-information setting (Fig. 3b). To explore how these results depend on the benefit of cooperation $b_1$ and on the selection strength $\beta$, Fig. 2d shows further simulations where we systematically vary both parameters. In all considered cases, state information is beneficial because it allows individuals to give more nuanced responses.

The second example has a similar transition vector as the first, with a single modification. This time, the inferior state is only left if at least one of the two players cooperates (Fig. 2e). Although this modification may appear minor, the resulting dynamics is strikingly different. We observe that with and without state information, individuals are now largely cooperative. However, they are most cooperative when individuals do not condition their strategies on the state information (Fig. 2f). For this stochastic game, we show in Supplementary Note 3 that already the traditional WSLS rule is subgame perfect for $2b_1 - b_2 \geq 2c$. As a result, WSLS is predominant in the no-information setting (Fig. 3d). In contrast, in the full-information setting, WSLS is subject to (almost) neutral drift by strategies that only differ from WSLS in a few bits (Fig. 3c). These other strategies may in turn give rise to the occasional invasion of defectors. Overall, we find that this stochastic game exhibits a benefit of ignorance when selection is sufficiently strong, and when cooperation is particularly valuable in the more profitable state (i.e., in the upper right corner of Fig. 2h).

These examples highlight three observations. First, just as there are instances in which state information is beneficial, there are also instances in which state information can reduce how much cooperation players achieve. Second, the stochastic games (transition vectors) for which state information is beneficial may only differ marginally from games with a benefit of ignorance. Finally, even if a stochastic game admits a benefit of ignorance, this benefit may not be present for all parameter values. Taken together, these observations suggest that in general, the effect of state information can be non-trivial and requires further investigation.

## A systematic analysis of the weak-selection limit

To explore more systematically in which cases there is a benefit of information (or ignorance), we study the class of all games with deterministic transition vectors. We first consider the limit of weak selection ($\beta \to 0$). Here, game payoffs only weakly influence how individuals adopt new strategies. While a vanishingly small selection strength is a mathematical idealization, this limit plays an important role in evolutionary game theory[43–45]. It often permits researchers to derive explicit solutions when analytical results are difficult to obtain otherwise. In our case, the limit of weak selection is particularly convenient, because it allows us to exploit certain symmetries between the two possible states $s_1$ and $s_2$, and between the two possible actions $C$ and $D$, see Supplementary Note 1. As a result, we show that instead of 64 stochastic games, we only need to analyze 24. For each of these 24 transition vectors $\mathbf{q}$, we explore whether information is beneficial, detrimental, or neutral (i.e., whether $V_0(\mathbf{q})$ is positive, negative, or zero).

First, we prove that half of the 64 stochastic games are neutral. In these games, the full-information and the no-information setting yield the same average cooperation rate in the limit of weak selection. Among the neutral games, we identify three (overlapping) subclasses. (*i*) The first subclass consists of those games that have an absorbing state (15 cases). Here, either the first or the second state can no longer be left once it is reached, because $q^1_{a\tilde{a}} = 1$ or $q^2_{a\tilde{a}} = 0$ for all $a$ and $\tilde{a}$. For these games, state information is neutral because players can be sure they are in the absorbing state eventually. (*ii*) In the second subclass, transitions are state-independent[31], which means $q^1_{a\tilde{a}} = q^2_{a\tilde{a}}$ for all $a$ and $\tilde{a}$ (6 additional cases). For deterministic transitions, state-independence implies that the current state can be directly inferred from the players' previous actions, even without obtaining explicit state information. (*iii*) In the third subclass, neutrality arises because of more abstract symmetry arguments, described in detail in Supplementary Note 1. In particular, while the games in the first two subclasses are neutral for all selection strengths, the games in the third subclass only become neutral for vanishing selection. One particular

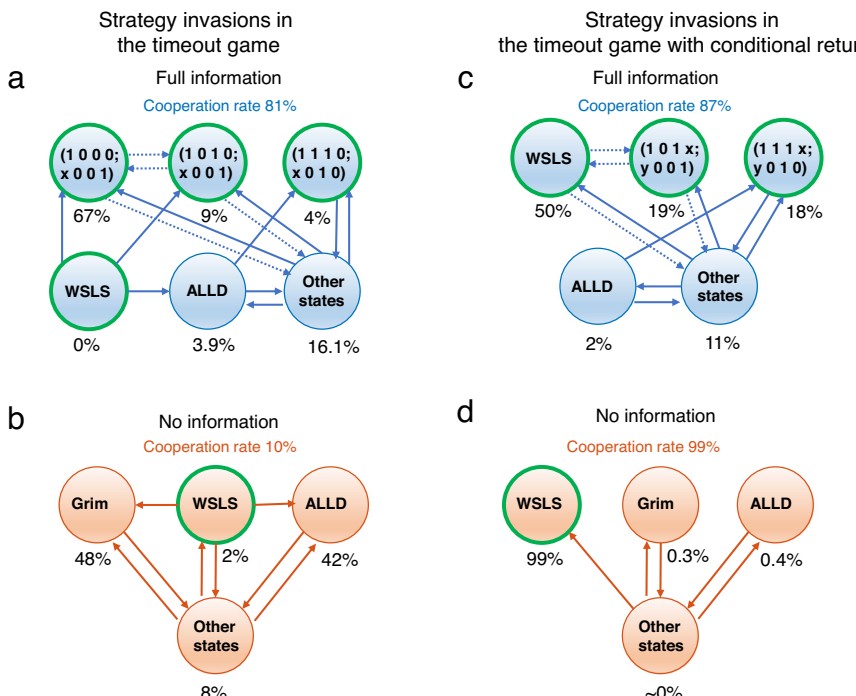

**Fig. 3 | Strategy invasion analysis for the two timeout games.** Here we analyze the invasion dynamics between different resident populations for the two examples considered in Fig. 2. Every circle represents a possible resident strategy. The frequency underneath indicates how often the respective resident population is observed over the course of the evolutionary process according to the invariant distribution. Strategies that have 100% self-cooperation rate (in the limit of rare errors) are highlighted with a green ring. Lines between the strategies represent the direction of selection. Solid lines indicate that the respective fixation probability is larger than $1/N$. Dotted lines indicate that the fixation probability is smaller than $1/N$

but greater than $1/(10N)$; in that case we speak of `almost-neutral drift'. **a**, **b** In the timeout game with full information, there are several highly cooperative strategies that are fairly robust against invasions. In contrast, for no information, players can only maintain cooperation with WSLS, which is unstable for the given parameter values. **c**, **d** The picture changes in the timeout game with conditional return. Here, WSLS is stable in the game with no information. In contrast, when there is full information, WSLS can be invaded through almost-neutral drift. Parameters are the same as in Fig. 2.

example of this last subclass is the game with transition vector $\mathbf{q} = (1, 0, 0; 1, 1, 0)$, which we studied in the previous section (Figs. 2e−h and 3c, d). There, we observed that this game can give rise to a benefit of ignorance when selection is intermediate or strong. Here, we conclude that this benefit disappears completely for vanishing selection (see also the lower boundary of Fig. 2h).

For the remaining 32 non-neutral cases, we identify a simple proxy variable that indicates whether or not the respective game exhibits a benefit of information for weak selection (Fig. 4a). Specifically, in a non-neutral game, information is beneficial if and only if $X > 0$, with $X$ being

$$X = \left( \mathbb{1}_{q_{CC}^1 = 1} + \mathbb{1}_{q_{CC}^2 = 0} \right) - \left( \mathbb{1}_{q_{DD}^1 = 1} + \mathbb{1}_{q_{DD}^2 = 0} \right). \qquad (4)$$

Here, $\mathbb{1}_A$ is an indicator function that is one if assertion $A$ is true and zero otherwise. One can interpret the variable $X$ as a measure for how easily the game can be absorbed in mutual cooperation ($X \geq 0$) or mutual defection ($X \leq 0$). For example, if a game has a transition vector with $q_{CC}^1 = 1$, groups can easily implement indefinite cooperation by choosing strategies with $p_{CC}^1 = 1$. By doing so, players ensure they remain in the first state, in which they again would continue to cooperate. Using the proxy variable $X$, we can conclude that there are two properties of transition vectors that make state information beneficial in the limit of weak selection. The transition vector either needs to allow players to coordinate on mutual cooperation in a stable environment ($q_{CC}^1 = 1$, $q_{CC}^2 = 0$); or it needs to prevent players from coordinating on mutual defection in a stable environment ($q_{DD}^1 \neq 1$, $q_{DD}^2 \neq 0$). Again by symmetry considerations, we find that there are as many games with a benefit of information as there are games with a benefit of ignorance (16 cases each, see Fig. 4a).

## Exploring the impact of other game parameters

After characterizing the case of weak selection, we next explore the dynamics under strictly positive selection. To this end, we numerically compute the population's average cooperation rate with and without state information, for each of the 64 stochastic games considered previously. To explore the impact of different game parameters, we systematically vary the strength of selection (Figs. 4b and S4), the benefit of cooperation (Figs. 4c and S5), and the error rate (Fig. S6). For 21 games, the evolving cooperation rates are the same with and without information. These games are neutral either because there is an absorbing state, or because transitions are state-independent (as described earlier). For the remaining cases, we find that a clear majority of them result in a benefit of information (Fig. 4b, c).

In the few cases with a consistent benefit of ignorance (the red squares in Figs. S4–S6), there is overall very little cooperation. As a result, the magnitude of this benefit is often negligible. Only in two cases one can find parameter combinations that lead to a sizeable benefit of ignorance. The first case is the stochastic game considered in Fig. 2e–h with transition vector $\mathbf{q} = (1, 0, 0; 1, 1, 0)$. The other case is a slight modification of the first, having the transition vector $\mathbf{q} = (1, 0, 1; 1, 1, 0)$. In both cases mutual cooperation leads to the more profitable first state. Moreover, in both cases, players can use WSLS to sustain cooperation even without state information, provided that $2b_1 - b_2 \geq 2c$. But even when this condition holds, the benefit of ignorance is constrained, because even fully informed populations tend to achieve substantial cooperation rates (Figs. S4–S6). Overall, these results suggest that for positive selection strengths, a sizeable benefit of ignorance is rare. Moreover, there seems to be no simple rule that predicts for which stochastic games we can expect a benefit of ignorance (see Supplementary Note 3, Section 3.3 for a more detailed discussion).

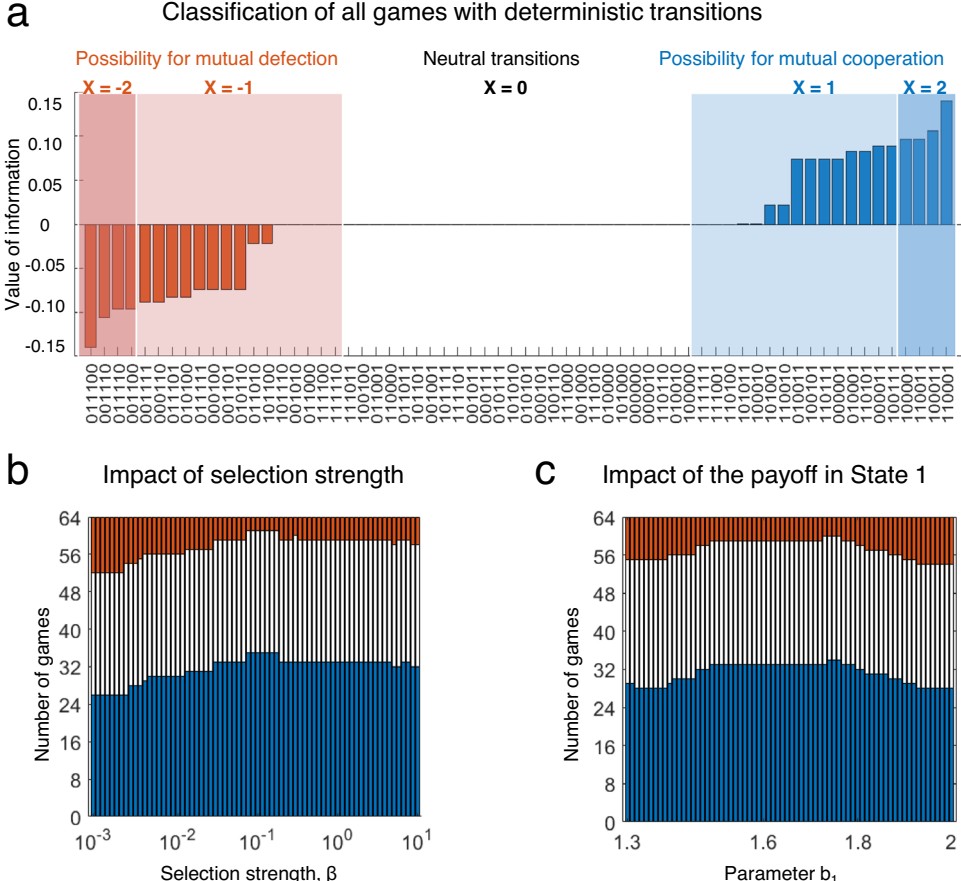

**Fig. 4 | Classification of games with deterministic transitions. a** In the limit of weak selection ($\beta \to 0$), we can define a simple proxy variable $X$ by Eq. (4) that indicates whether information is better, neutral, or worse for games with deterministic transitions. When $X = 0$ or when the stochastic game has an absorbing state, the game is neutral. In all other cases, there is either a benefit of information (when $X > 0$) or a benefit of ignorance (when $X < 0$). The bar diagram depicts the respective value of information for each of the 64 possible cases. The panel is symmetric; for each game with a benefit of information, there is a corresponding game with the same benefit of ignorance. **b** Once we increase the selection strength, games with a benefit of information become predominant. **c** We also study the effect of $b_1$ (the benefit of cooperation in the more profitable state) under strong selection, $\beta = 10$. Again, most games are either neutral or show a benefit of information. Unless explicitly noted otherwise, we use the same parameters as before.

## The effect of environmental stochasticity

In our analysis so far, we assumed that the environment changes deterministically. Individuals who know the present state and the players' actions can therefore anticipate the game's next state. This form of predictability may overall diminish the impact of explicit state information because it reduces uncertainty. In the following, we extend our analysis to allow for stochasticity in the game's transitions. To gain some intuition, we start with a simple example taken from the previous literature[31] (see Fig. 5a for a depiction). According to the game's transition vector, $\mathbf{q} = (1, 0, 0, q, 0, 0)$, players always find themselves in the less profitable second state if one or both players defect. If both players cooperate, however, they either remain in the first state (if they are already there), or they transition to the first state with probability $q$ (if they start out in the second state). This stochastic game represents a scenario in which an environment deteriorates immediately once players defect. If players resume to cooperate, it may take several rounds for the environment to recover.

For this example, we find that the value of information varies non-trivially, depending on the transition probability $q$ and the strength of selection $\beta$ (Fig. 5b–e). Overall, parameter regions with a benefit of ignorance seem to prevail (Fig. 5f). To obtain analytical results, again we study the game for weak selection ($\beta \to 0$). In that case, the value of information can be computed explicitly, as $V_0(\mathbf{q}) = -\frac{3q(1-q)}{64(1+q)}$. In particular, there is a benefit of ignorance for all intermediate values $q \in (0, 1)$. This benefit becomes most pronounced for $q = \sqrt{2} - 1$ (for

more details, see Supplementary Note 3, Section 3.4). As we increase the selection strength, however, the dynamics can change, depending on $q$. For small $q$, we continue to observe a benefit of ignorance, whereas for larger $q$ information tends to become beneficial (Fig. 5f).

To explore the scenarios with a benefit of ignorance, we record which strategies players adopt for $q = 0.2$. Without state information, we find that players adopt WSLS almost all of the time (Fig. 5g). In contrast, when players condition their strategies on state information, WSLS is risk-dominated by a strategy that has been termed Ambitious WSLS[31] (AWSLS). AWSLS differs from WSLS after mutual cooperation, in which case AWSLS only cooperates when players are in the first state (i.e., $q_{CC}^1 = 1$ but $q_{CC}^2 = 0$). Once AWSLS is common in the population, it opens up opportunities for less cooperative strategies to invade. In particular, also non-cooperative strategies like Always Defect (ALLD) are adopted for a non-negligible fraction of time (Fig. 5h). Overall, we find that predicting the effect of information is non-trivial. While some parameter combinations favor populations with full information, we also observe a benefit of ignorance for a significant portion of the parameter space.

To obtain a more comprehensive picture, we numerically analyze all stochastic games with single-stochastic transition vectors. Because the corresponding transition vectors have exactly one entry $q$ between 0 and 1, there are $6 \cdot 2^5 = 192$ cases in total. We find several regularities. First, similarly to games with deterministic transitions, we find that there are 24 transition vectors for which the game is neutral. In all of

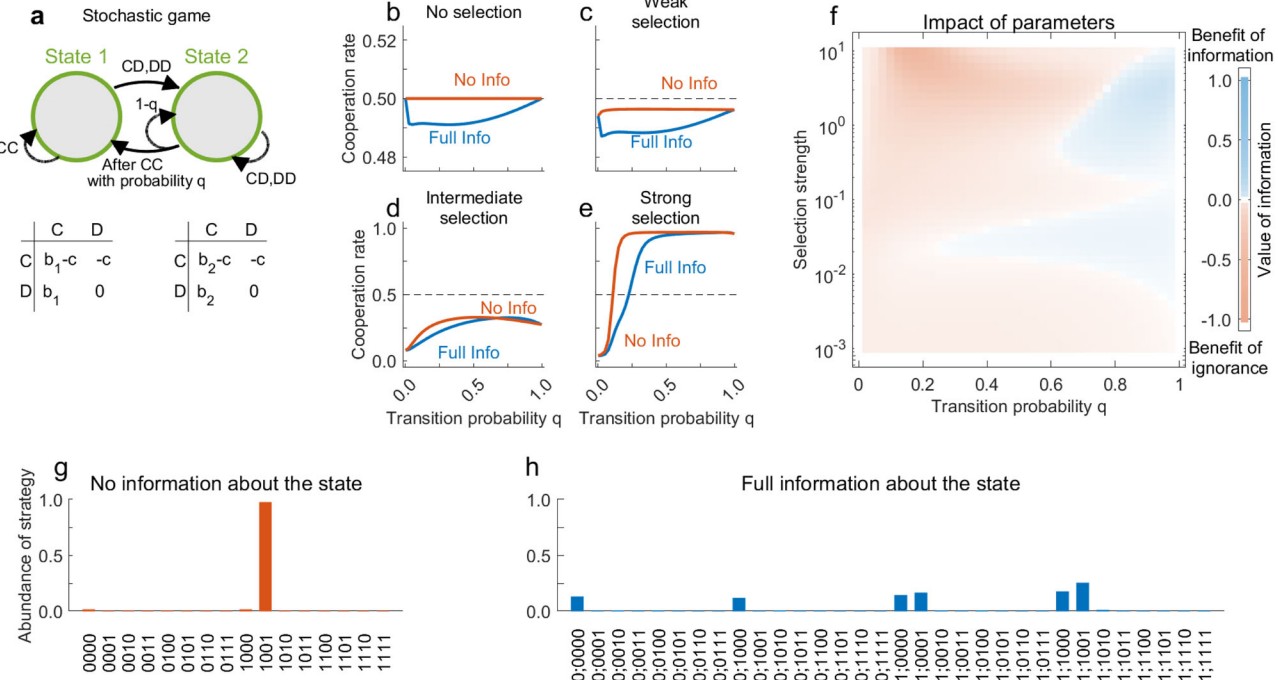

**Fig. 5 | Benefit of ignorance in a game with a single-stochastic transition. a** We consider a stochastic game in which defection by any player leads to the inferior second state. From there, players return to the more profitable first state after mutual cooperation with probability $q$. **b–f** We compute numerically exact cooperation rates for the stochastic game with no information and with full information, for different values of the transition probability $q$ and selection strength $\beta$. For no and weak selection, there is a benefit of ignorance for all values of $q \in (0, 1)$. For intermediate and strong selection, a benefit of ignorance persists when the transition probability $q$ is sufficiently small. **g,h,** We plot how often each strategy is played for $q = 0.2$ and strong selection. Because any defection leads to state 2, we can use a simplified notation for full-information strategies, $\mathbf{p} = (p_{CC}^1; p_{CC}^2, p_{CD}^2, p_{DC}^2, p_{DD}^2)$; the remaining three entries $p_{CD}^1, p_{DC}^1, p_{DD}^1$ are irrelevant (Section 3.4 in Supplementary Note 3). We observe that when there is no information, most players adopt WSLS. With full information, there is no clearly winning strategy. Baseline parameters are the same as before. For no, weak, intermediate and strong selection we use $\beta = 0$, $\beta = 0.001$, $\beta = 1$, and $\beta = 10$, respectively.

these games, one of the two states is absorbing. Second, we analyze the remaining cases in the limit of vanishing selection (Fig. S7). Most of these games follow the rule defined by the proxy variable $X$ in Eq. (4), with some exceptions discussed in detail in Supplementary Note 2. Finally, for positive selection strengths we can again compute the players' average cooperation rates numerically. We do this for all 192 families of games for weak (Fig. S8), intermediate (Fig. S9), and strong selection (Fig. S10). Similar to the case of deterministic transitions, state information is beneficial in an absolute majority of cases (Fig. S11). However, exceptions can and do occur. A notable benefit of ignorance arises most frequently when mutual cooperation in the more beneficial state leads the players to remain in that state, and when mutual defection in any state is punished with deteriorating environmental conditions.

Our computational methods are not limited to games with deterministic or single-stochastic transitions. To obtain a comprehensive understanding of the general effect of state information, we systematically explore the space of all stochastic transition vectors. To make this analysis feasible, we assume the entries of **q** are taken from a finite grid $q_{ij}^k \in \{0, 0.2, 0.4, 0.6, 0.8, 1.0\}$, leading to $6^6 = 46,656$ possible cases. Our numerical results again confirm that for the majority of these cases, environmental information is beneficial (Fig. S12b). Although there is also a non-negligible number of games for which populations are better off without information, the respective benefit of ignorance is often small (Fig. S12a).

## Discussion

When people interact in a social dilemma, their actions often have spillovers to their social, natural, and economic environment[56–59]. Changes in the environment may in turn modulate the characteristics of the social dilemma. One important example of such a feedback loop

is the 'tragedy of the commons'[60]. Here, groups with little cooperation may deteriorate their environment, thereby restricting their own feasible long-run payoffs.

Such spillovers between the groups' behavior and their environment can be formalized as a stochastic game[28]. In stochastic games, individuals interact for many time periods. In each period, they may face a different kind of social dilemma (state). The way they act in one state may affect the state they experience next. Recently, stochastic games have become a valuable model for the evolution of cooperation, because changing environments can reinforce reciprocity[31–33]. In particular, the evolution of cooperation may be favored in stochastic games even if cooperation is disfavored in each individual state[31], see also Fig. S2a, b. However, implicit in these studies is the assumption that individuals are perfectly aware of the state they are in. Here, we systematically explore the implications of this assumption. We study to which extent individuals learn to cooperate, depending on whether or not they know the present state of their environment. We say the stochastic game shows a benefit of information if well-informed groups tend to be more cooperative. Otherwise, we speak of a benefit of ignorance.

Already for the most basic instantiation of a stochastic game, with two individuals and two states, we find that the impact of information is non-trivial. All three cases are possible: state information can be beneficial, neutral, or detrimental for cooperation. To explore this complex dynamics, we employ a mixture of analytical techniques and numerical approaches. Analytical results are feasible in the limiting case of weak selection[43–45]. Here, we observe an interesting symmetry. For every stochastic game in which there is a benefit of information, there is a corresponding game with a benefit of ignorance. This symmetry breaks down for positive selection. As selection increases, we observe more and more cases in which state information becomes

beneficial. Moreover, in those few cases in which a benefit of ignorance persists, this benefit tends to be small. These results highlight the importance of accurate state information for responsible decision making.

However, our research also highlights a few notable exceptions. We identify several ecologically plausible scenarios where individuals cooperate more when they ignore their environment's state. One example is the game displayed in Fig. 2e–h. Here, players only remain in the profitable state when they both cooperate. Once they defect, they transition to the inferior state. From there, they can only escape if at least one player cooperates. This game reflects a scenario where the group's environment reinforces cooperation. Cooperative groups are rewarded by maintaining access to the more profitable state. Non-cooperative groups are punished by transitioning to an inferior state. For this kind of environmental feedback it was previously observed that the simple WSLS strategy can sustain cooperation easily[31–33]. WSLS can be instantiated without any state information. Once a population settles at WSLS, providing state information can even be harmful; in that case, individuals may deviate towards more nuanced strategies, which in turn can destabilize cooperation. In this sense, our study mirrors previous results suggesting that richer strategy spaces can sometimes reduce a population's potential to cooperate[61].

To allow for a systematic treatment, we focus on comparably simple games. Nevertheless, the number of games we consider is huge. For example, if all transitions between states are assumed to be deterministic (independent of chance), there are 64 cases to consider (Figs. S4–S6). If all but one transition are deterministic, we obtain 192 families of games (each having a free parameter $q \in [0, 1]$, Figs. S7–S10). In addition, we also systematically explore the set of fully stochastic transition functions, by considering 46,656 different cases (Fig. S12). In all these instances, we observe that seemingly innocent changes in the environmental feedback or in the game parameters can lead to complex changes in the dynamics. In particular, games with a benefit of information may turn into games with a benefit of ignorance. As shown in Fig. S13, we observe a similar sensitivity in games with more than two players. These observations suggest that there may be no simple rule that predicts the impact of state information. These difficulties are likely to further increase as we extend the model to more complex strategies[47–52], or environments with multiple states[31].

Overall, we believe our work makes at least two contributions. First, we introduce a simple and easily generalizable framework to explore how state information (or the lack thereof) affects the evolution of cooperation. This framework can be generalized into various directions. For example, in our model we compare two limiting cases. We either consider a population in which no one knows the state of the environment, or in which everyone gets precise information about the environment's state. There are many interesting cases in between. In some applications, population members may only obtain an imperfect signal of the environment's true state[42]. Alternatively, one may adapt our model to explore games with information asymmetries. As one instance of such a model extension, individuals may choose to acquire state information at a small cost. Such a model would allow researchers to explore whether individuals acquire information exactly in those games for which we find a benefit of information.

As our second contribution, our results illustrate the intricate dynamics that arise in the presence of environmental, informational, and behavioral feedbacks. By exploring these feedbacks in elementary stochastic games, we can better understand the more complex dynamics of the socio-ecological systems around us.

## Methods

### Calculation of payoffs in stochastic games

In this study, we compare the evolutionary dynamics for two strategy sets. The first set $\mathcal{S}_F$ is the set of all memory-one strategies for the full-information setting. The second set $\mathcal{S}_N$ consists of all memory-one

strategies for the no-information setting. Equivalently, we can define $\mathcal{S}_N$ as the set of all full-information strategies that do not condition their behavior on the current state,

$$\mathcal{S}_N = \left\{ \mathbf{p} \in \mathcal{S}_F \mid p^1_{a\tilde{a}} = p^2_{a\tilde{a}} \; \forall a, \tilde{a} \in \{C, D\} \right\}. \tag{5}$$

We denote by $\mathcal{P}_F$ and $\mathcal{P}_N$ the respective sets of deterministic strategies, for which all entries are required to be either zero or one. In the following, we describe how to calculate payoffs when players have full information. Since any strategy for the case of no information can be associated with a full-information strategy, the same method also applies to the case of no information.

As our baseline, we consider games that are infinitely repeated and in which there is no discounting of the future. Given player 1's effective memory-1 strategy $\mathbf{p}$ and player 2's effective strategy $\tilde{\mathbf{p}}$, such games can be described as a Markov chain. The states of this Markov chain correspond to the eight possible outcomes $\omega = (s_i, a, \tilde{a})$ of a given round. Here, $s_i \in \{s_1, s_2\}$ reflects the environmental state, and $a, \tilde{a} \in \{C, D\}$ are player 1's and player 2's actions, respectively. The transition probability to move from state $\omega = (s_i, a, \tilde{a})$ in one round to $\omega' = (s'_i, a', \tilde{a}')$ in the next round is a product of three factors,

$$m_{\omega, \omega'} = x \cdot y \cdot \tilde{y}. \tag{6}$$

The first factor

$$x = \begin{cases} q^i_{a\tilde{a}} & \text{if } s'_i = s_1 \\ 1 - q^i_{a\tilde{a}} & \text{if } s'_i = s_2 \end{cases} \tag{7}$$

reflects the probability to move from environmental state $s_i$ to $s'_i$, given the player's previous actions. Since the game is symmetric, we note that $q^i_{DC}$ is defined to be equal to $q^i_{CD}$. The other two factors are

$$y = \begin{cases} p^{i'}_{a\tilde{a}} & \text{if } a' = C \\ 1 - p^{i'}_{a\tilde{a}} & \text{if } a' = D \end{cases}' \tag{8}$$

$$\tilde{y} = \begin{cases} \tilde{p}^{i'}_{\tilde{a}a} & \text{if } \tilde{a}' = C \\ 1 - \tilde{p}^{i'}_{\tilde{a}a} & \text{if } \tilde{a}' = D. \end{cases} \tag{9}$$

They correspond to the conditional probability that each of the two players chooses the action prescribed in $\omega'$. By collecting all these transition probabilities, we obtain an $8 \times 8$ transition matrix $M = (m_{\omega, \omega'})$. Assuming that players are subject to errors and that the game's transition vector satisfies $\mathbf{q} \neq (1, 1, 1, 0, 0, 0)$, this transition matrix has a unique left eigenvector $\mathbf{v}$. The entries $v^i_{a\tilde{a}}$ of this eigenvector give the frequency with which players observe the outcome $\omega = (s_i, a, \tilde{a})$ over the course of the game. For a given transition vector $\mathbf{q}$, we can thus compute the first players' expected payoff as

$$\pi(\mathbf{p}, \tilde{\mathbf{p}}) = b_1 (v^1_{CC} + v^1_{DC}) + b_2 (v^2_{CC} + v^2_{DC}) - c (v^1_{CC} + v^1_{CD} + v^2_{CC} + v^2_{CD}) \tag{10}$$

The second player's payoff can be computed analogously. Similarly, the average cooperation rate of the two players can be defined as follows.

$$\gamma(\mathbf{p}, \tilde{\mathbf{p}}) = \left( v^1_{CC} + \frac{v^1_{CD} + v^1_{DC}}{2} \right) + \left( v^2_{CC} + \frac{v^2_{CD} + v^2_{DC}}{2} \right). \tag{11}$$

In this work, we focus on games without discounting. However, similar methods can be applied to games in which future payoffs are discounted by a factor of $\delta$ (or equivalently, to games with a continuation probability $\delta$). For $\delta < 1$, instead of computing the left

eigenvector of the transition matrix, we define $\mathbf{v}$ to be the vector

$$\mathbf{v} = (1-\delta)\mathbf{v_0}\sum_{t=0}^{\infty}(\delta M)^t = (1-\delta)\mathbf{v_0}(I_8 - \delta M)^{-1}. \quad (12)$$

In this expression, $\mathbf{v_0}$ is the vector that contains the probabilities to observe each of the eight possible states $\omega$ in the very first round. Moreover, $I_8$ is the $8 \times 8$ identity matrix. Similar to before, the entries of $\mathbf{v} = (v_{a,\tilde{a}}^i)$ represent the weighted average that describes how often the two players visit the state $\omega$ over the course of the game[62]. The payoffs and the average cooperation rate can then again be computed with the formulas in (10) and (11). We use this approach when we explore the impact of the continuation probability $\delta$ on the robustness of our results in Fig. S1e, f.

## Evolutionary dynamics

To model how players learn to adopt new strategies over time, we study a pairwise comparison process[55] in the limit of rare mutations[63–66]. We consider a population of fixed size $N$. Initially, all players adopt the same resident strategy $\mathbf{p}_R = ALLD$. Then one of the players switches to a randomly chosen alternative strategy $\mathbf{p}_M$. This mutant strategy may either go extinct or reach fixation, depending on which payoff it yields compared to the resident strategy. If the number of players adopting the mutant strategy is given by $k$, the expected payoffs of the two strategies is

$$\pi_R(k) = \frac{N-k-1}{N-1} \cdot \pi(\mathbf{p}_R, \mathbf{p}_R) + \frac{k}{N-1} \cdot \pi(\mathbf{p}_R, \mathbf{p}_M), \quad (13)$$

$$\pi_M(k) = \frac{N-k}{N-1} \cdot \pi(\mathbf{p}_M, \mathbf{p}_R) + \frac{k-1}{N-1} \cdot \pi(\mathbf{p}_M, \mathbf{p}_M) \quad (14)$$

Based on these payoffs, the fixation probability of the mutant strategy can be computed explicitly[43,67],

$$\rho(\mathbf{p}_R, \mathbf{p}_M) = \frac{1}{1 + \sum_{i=1}^{N-1}\prod_{k=1}^{i}\exp\left[-\beta\left(\pi_M(k) - \pi_R(k)\right)\right]}. \quad (15)$$

As the selection strength parameter $\beta$ approaches zero, this fixation probability approaches the neutral probability $1/N$, as one may expect. As $\beta$ increases, the fixation probability is increasingly biased in favor of mutant strategies with a high relative payoff.

If the mutant fixes, it becomes the new resident strategy. Then another mutant strategy is introduced and either fixes or goes extinct. By iterating this basic process for $\tau$ time steps, we obtain a sequence $(\mathbf{p}_0, \mathbf{p}_1, \mathbf{p}_2, ..., \mathbf{p}_\tau)$ where $\mathbf{p}_t$ is the resident strategy present in the population after $t$ mutant strategies have been introduced. Based on this sequence, we can calculate the population's average cooperation rate and payoff as

$$\hat{\pi} = \lim_{\tau \to \infty}\frac{1}{\tau+1}\sum_{t=0}^{\tau}\pi(\mathbf{p}_t, \mathbf{p}_t), \quad (16)$$

$$\hat{\gamma} = \lim_{\tau \to \infty}\frac{1}{\tau+1}\sum_{t=0}^{\tau}\gamma(\mathbf{p}_t, \mathbf{p}_t). \quad (17)$$

Because the evolutionary process is ergodic for any finite $\beta$, these time averages exist and are independent of the population's initial composition.

If players have infinitely many strategies, the payoff and cooperation averages in (16) can only be approximated, by simulating the above described process for a sufficiently long time $\tau$. However, when strategies are taken from a finite set $\mathcal{P}$, these quantities can be computed exactly. In that case, the evolutionary dynamics can again be described as a Markov chain[63]. Each state of this Markov chain corresponds to one possible resident population $\mathbf{p} \in \mathcal{P}$. Given that the current resident population uses $\mathbf{p}$, the probability that the next resident population uses strategy $\tilde{\mathbf{p}} \neq \mathbf{p}$ is given by $\rho(\mathbf{p}, \tilde{\mathbf{p}})/|\mathcal{P}|$. By calculating the invariant distribution $\mathbf{w} = (w_\mathbf{p})$ of this Markov chain, we can compute the average cooperation rates and payoffs according to Eq. (16) by evaluating

$$\hat{\pi} = \sum_{\mathbf{p} \in \mathcal{P}} w_\mathbf{p} \cdot \pi(\mathbf{p}, \mathbf{p}), \quad (18)$$

$$\hat{\gamma} = \sum_{\mathbf{p} \in \mathcal{P}} w_\mathbf{p} \cdot \gamma(\mathbf{p}, \mathbf{p}). \quad (19)$$

Herein, we perform these calculations for the specific strategy sets for full information and no information, $\mathcal{P}_F$ and $\mathcal{P}_N$, respectively. By comparing the respective averages $\hat{\gamma}^F$ and $\hat{\gamma}^N$, we characterize for which stochastic games there is a benefit of information, by computing $V_\beta(\mathbf{q}) = \hat{\gamma}^F - \hat{\gamma}^N$.

We use this process based on deterministic strategies, pairwise comparisons, and rare mutations for all of our main text figures. As robustness checks, we present several variations of this model in the SI. For example, in Fig. S1a,b, we show simulation results for players with stochastic memory-1 strategies. To this end, we assume that mutant strategies are randomly drawn from the spaces $\mathcal{S}_F$ and $\mathcal{S}_N$. To make sure that strategies close to the corners get sufficient weight, the entries $p_{a\tilde{a}}^i$ are sampled according to an arcsine distribution, as for example in Nowak and Sigmund[12]. Similarly, in Fig. S1h, i, we show simulations for positive mutation rates. In Fig. S2a, b, we compare the results from Fig. 2 to a setup in which players only engage in the game in the first state (without any transitions), or in which they only engage in the game in the second state. In addition, in Fig. S2c, d, we run simulations when players are unable to condition their behavior on the outcome of the previous round. Finally, to explore whether our qualitative results depend on the specific learning process we use, we have also implemented simulations with an alternative learning process, introspection dynamics[68–70]. The respective results are shown in Fig. S3.

## Reporting summary

Further information on research design is available in the Nature Portfolio Reporting Summary linked to this article.

## Data availability

The generated simulation data is available at https://github.com/kleshnina/stochgames_info.

## Code availability

All numerical computations were performed with Matlab. For some of the symbolic calculations we used Mathematica. The respective code is available at zenodo[71] and on GitHub: https://github.com/kleshnina/stochgames_info.

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

## Acknowledgements
This work was supported by the European Research Council CoG 863818 (ForM-SMArt) (to K.C.), the European Research Council Starting Grant 850529: E-DIRECT (to C.H.), the European Union's Horizon 2020 research and innovation program under the Marie Sklodowska-Curie Grant Agreement #754411 and the French Agence Nationale de la Recherche (under the Investissement d'Avenir programme, ANR-17-EURE-0010) (to M.K.).

## Author contributions
All authors conceived and discussed the study; S.S. ran some preliminary simulations; M.K. and C.H. analyzed the model, conducted further simulations, and wrote the first draft of the manuscript; M.K., C.H., S.S., M.N. and K.C. discussed the results and edited the manuscript.

## Competing interests
The authors declare no competing interests.
