## [Peer Review File · Nature Communications]

The effect of environmental information on evolution of cooperation in stochastic gamesReviewers' Comments:

Reviewer #1:

Remarks to the Author:

This fascinating paper studies the effect of environmental information on the degree of cooperation that evolves in repeated games, in a scenario where there is switching between payoff matrices in a manner that depends on the behavior of the participants in the game. This is a natural extension of previous work on iterated games and environmental feedback, and the authors find counterintuitively that having access to environmental information does not necessarily result in higher levels of cooperation. They systematically classify all the "deterministic stochastic games" (i.e. stochastic games for which the probability of switching between payoff matrices is 1 or 0 depending on the actions of the interacting players in the previous round) as benefitting from the availability of environmental information or not, using a mixture of analytical results (under weak selection and weak mutation), and via simulation (under stronger selection). This work is a valuable contribution to the study of cooperation and has implications that will be of relevance to readers from a wide variety of disciplines. I therefore support publication following revision.

Detailed comments

1) The term "deterministic stochastic games" is a bit odd. Indeed the title refers to "stochastic games" but the authors mainly study deterministic switching. I appreciate the use of the term stochastic is consistent with what has come before, but you might want to rethink having it in the title.

2) Only deterministic memory-1 strategies are studied. It's unclear to me whether allowing non-deterministic strategies would disrupt the results i.e. are the results for the timeout game with conditional return critically dependent on the interaction between the deterministic switching and the deterministic strategies? Another perhaps easier way to get at this would be to systematically vary the error rate in the player's strategies. I appreciate that Fig. 5 looks at an almost-deterministic game and finds qualitatively similar results to Fig. 2, but it's not clear to me how the system will behave when behavior is not highly deterministic.

3) When there is a cost to information, do non-informed strategies outcompete informed strategies i.e. does ignorance evolve?

4) We have found (Stewart, Parsons and Plotkin, PNAS, 2016) that when we compare strategies that have little choice (cooperate or defect) to strategies that have lots of choice (choose how much to cooperate between 0 and full) that there can be a cost to choice, depending on the returns for cooperation. The intuition for this deleterious effect of choice is that increasing choice effectively generates sub-optimal fitness peaks, which the population can get stuck on. It's not necessary to address our paper specifically, but it seems likely a similar interpretation can be used for the results here and it might be worth noting.

5) It's not at all clear that the indicator function Eq. 4 is represented correctly in Figure 4a. Presumably the x-axis in Fig. 4a is the transition vector q ? Why then are $(1,1,1;1,1,1)$ and $(0,0,0;0,0,0)$ at the right hand end of the x-axis not classified as neutral? Clearly information is not beneficial in either of these cases and indeed the indicator function for $(1,1,1;1,1,1)$ should be $X=(1+0)-(1+0)=0$. I suspect some axis mislabelling has occurred.

6) The panel labelling in Figure 2 is also messed up

Alex Stewart

Reviewer #2:

Remarks to the Author:

The manuscript studies the effect of environment information on emergence of cooperation using the stochastic games framework. In particular, the framework considers two players in a stochastic game with two actions (Cooperate, Defect) and two states (profitable, less profitable for cooperation). The evolutionary behavior follows a smoothed best response with some noise (error) in choosing among possible memory-1 policies. In the full information case, players observe the current state and performance of other strategies in the last step. In the no information case, players only observe the performance of other strategies in the last step, but they cannot condition their current policy on the state of the environment. The paper provides a systematic analysis of the arising cooperation levels in full and no environment information settings considering the benefit of cooperation in the profitable state, selection strength, and transition probabilities. The results point to a preference for environment information (full information) in emergence of cooperation. Specific scenarios in which no environmental information leads to higher cooperation levels are rare, and tend to have limited improvement in cooperation levels over the full information settings.

The paper considers the emergence of cooperation in populations by incorporating relatively unexplored features in prior works: environment feedback, and information availability. The framework design is simple, yet expectedly exhibit rich set of dynamics. Most of the results are obtained by an exhaustive simulation of the entire set of outcomes. A few of the results rely on analytical derivations. Other analytical results simplify the search space, easing the computational burden of exhaustive simulation. The paper is well-written with a clear and relevant motive. The take-away results are often mixed, in the sense that there is no clear winner between the two options (full or no information). Overall, the paper takes a systematic approach to understanding the emergence of cooperation in relatively unexplored settings.

1) My main comment is that there is no guarantee provided for the convergence of the learning dynamics in the proposed setting. Without such a guarantee, the outcomes from the simulations and interpretations could depend on the behavior dynamics. Indeed from Fig. 3, it seems the population can shift between different policies. I.e., there exists a set of policies that are not eliminated by the selection procedure. In this case, the relation between the emergence of cooperation and the amount of time spent in the preferred state can become tenuous. I realize the analysis focuses on the relative cooperation rates for populations with full and no information, but a note on the convergence properties of the learning dynamics and its relation to surviving policies is necessary.

2) There are other learning dynamics that can be considered. Should we expect the results to change as a the learning dynamics change? In other words, are the results robust to changes in the learning dynamics?

3) In the SI, there is a mention on how we can rely on a subset of the policies available at the full information setting to study the no information setting. This is intuitive, since some of the policies in the full information can choose to ignore state information, and thus mimic the no state information setting. Perhaps this property can be used to make statements about asymmetric information where one set of agents is given the state information and the other set is not.

4) A major modeling assumption is that there is no discounting of future payoffs. The discounting could be a significant factor in determining the relative benefit of information. This assumption and its potential implications should be clearly specified and discussed in text.

5) Proposition 3's proof is by Mathematica which is not a valid argument/proof.

6) Many times the reader is referred to the SI within the text. However, it is not clear which part of SI is related. The in-text referrals to the SI should be more specific. It should specify which equation/result/figure the reader should see for ease of following the flow.

Some other minor comments follow:

7) There is a significant overlap between SI and methods. In particular, SI Sections 1.2 - 1.4 and parts of Section 2 appear in methods verbatim. There is a lot of redundancy.

8) Lemma 1's statement is not easy to follow given that it requires flipping through several pages before to be reminded of the definitions. Either remind the reader of the definitions within the lemma's statement or in text right before the lemma's statement.

9) There is a reference to Fig. 4d which does not exist.

10) Please make sure all acronyms, e.g., ALLD, TFT, are defined the first time they are used.

Reviewer #3:

Remarks to the Author:

****Summary:**** The authors investigate the influence of state information on the evolution of cooperation in stochastic games. Specifically, given the environmental transition function, they investigate whether memory-1 strategies that condition on the environmental state (i.e., taking state information into account) are better, neutral, or worse than a state-unconditional memory-1 strategy. Their evolutionary model consists of the following parameters: the selection strength β , the mutation rate μ , and the strategy-error rate ϵ . The manuscript provides an extensive analysis (combining analytical with simulation results) of deterministic state transitions in coupled two-state, two-agent, two-action Prisoner's dilemmas. Extensions to stochastic transition functions and three-agents public good games are also considered. Overall, the authors find the influence of state information on the evolution of cooperation in stochastic games to be non-trivial, i.e., it can be beneficial, neutral, and harmful, with non-trivial dependence on the model's parameters.

****Overall assessment:**** Overall, I enjoyed the paper, which makes a relevant contribution. It is well written and provides an extensive analysis that robustly underpins the main finding of the work. Yet, I believe some improvement can be made. These concern a clarification and contextualization of the model and its main findings (Comments 1-5) and some clarifications and suggestions regarding specific aspects (Comments 6-8).

****Specific comments****

1) In its current form, the main take-home message is that the effect of state information on the evolution of cooperation in stochastic games is non-trivial. While this is an important finding, my impression is that more can be said from all the analyses. The work would benefit from distilling its non-trivial analyses and results into relatable conclusions. For example, I would be excited to see a summary of the conditions when ignorance promotes the evolution of cooperation in stochastic games. And when it does, when is it even better than conditional strategies? Currently, the authors only give one example (Fig. 2e-h). But from Figs. S1-S3, I believe, games 51-55 are also possible candidates, among others. What can be said about the similarities between their transition function? How do the evolutionary parameters β , μ , ϵ affect the outcomes?

2) What are the consequences of this work? Despite the conceptual nature of this model, it would be interesting to the readers to hear about possible consequences the authors envision from their study. Be it a theoretical advancement on the evolution of cooperation in stochastic games, a possible

explanation why ignorance (about the state information) exists, or eventually looking for cooperation-promoting mechanisms that utilize the form of ignorance studied in this work.

3) Regarding the appropriate referencing to previous literature, the manuscript would benefit from a discussion on how the presented approaches relate to

1) the value of information in economics, e.g.,

- Levine, P., & Ponsard, J. P. (1977). The values of information in some nonzero-sum games. *International Journal of Game Theory*, 6(4), 221-229.

- Bagh, A., & Kusunose, Y. (2020). On the economic value of signals. *The BE Journal of Theoretical Economics*, 20(1).

2) the framework of partial observability, e.g.,

- Hansen, E. A., Bernstein, D. S., & Zilberstein, S. (2004, July). Dynamic programming for partially observable stochastic games. In *AAAI* (Vol. 4, pp. 709-715).

- Barfuss, W., & Mann, R. P. (2022). Modeling the effects of environmental and perceptual uncertainty using deterministic reinforcement learning dynamics with partial observability. *Physical Review E*, 105(3), 034409.

4) The authors study the evolution of cooperation of memory-1 strategies either with or without additional conditioning on the environmental states. Doing so enables two possible reasons for the evolution of cooperation, conditioning on the environment and conditioning on the previous actions. To better contextualize the results, it would be helpful to compare them to two baselines. First, conditional memory-zero strategies, which only condition on the environmental state, might evolve to cooperate. Second, how do memory-one strategies evolve in the static game without environmental transitions?

5) The authors provide an extensive mathematical analysis. However, sometimes the manuscript reads as if some modeling assumptions have only been taken to be able to execute some mathematical techniques. For example, the manuscript investigates the weak selection limit multiple times, as often done in evolutionary analysis. But I would welcome a justification or reasoning for why this is interesting, other than analytical tractability. For example, the yellow boxes in Fig. S1 clearly show that the results of the weak selection limit do not necessarily carry over to non-zero selection strengths. Another modeling assumption that is introduced rather ad-hoc is the error rate ϵ . I believe the authors have done so to avoid non-ergodic Markov chains when calculating the payoffs. While this is a valid reason, what modeling reasons apply, such as the agents' imperfection in carrying out a deterministic strategy or the need for exploration as frequently considered as the exploration-exploitation trade-off in reinforcement learning?

6) It did not become clear to me why the authors chose to study almost-deterministic transition functions. What is special or interesting about this type?

7) Something must have gone wrong with the labeling of Fig. 4. I see only three subfigures, a, b, and c, while the caption refers to four: a, b, c, and d. I believe Eq. 4 was part of the Figure at some point. Also, the caption has a different symbol for the value of information than the main text. Furthermore, I'm unable to replicate the relationship between the transition functions (as the labels on the x-axis) and the value of X (Eq. 4). For example, let's focus on $X=2$, which can only be achieved if the transition function $(q1_{CC}, q1_{CD}, q1_{DD}, q2_{CC}, q2_{CD}, q2_{DD})$ as given in Eq. 1 is $(1_{00}, 1)$ with either 0 or 1. But the four transition functions in Fig. 4a associated with $X=2$ are $(111101, 111110, 111111, 000000)$. Finally, in the text, Eq. 4 is introduced to apply to the 32 non-neutral cases. However, Fig. 4a shows all 64 cases, with the neutral transitions displayed as $X=0$. This suggests that Eq. 4 actually applies to all 64 cases.

8) For the large table figures in the supporting information, it would be helpful if they were ordered according to the classes introduced in the main text: neutral (absorbing states, state-independent

transitions, symmetries) and non-neutral ($X > 0$, $X < 0$).

First of all, we would like to thank all reviewers for their comments and suggestions on how to improve our manuscript. Their feedback was extremely constructive and helpful. In the meantime, we have addressed all their suggestions. Below we briefly summarize the major changes that we made to the manuscript as a response:

- 1) To address comments made by all three reviewers, we have run additional simulations to further explore the robustness of our findings. The first set of simulations explores the impact of stochastic strategies, errors, discounted games, and of games with abundant mutations. The respective results are summarized in the new **Supplementary Figure 1**. With the second set of simulations, we provide a control scenario for the results we present in the main text. For this control, we explore the dynamics when players act in a fixed environment, or when they are restricted to use strategies that only depend on the current state of the environment. The respective simulation results are shown in **Supplementary Figure 2**. Finally, with the third set of simulations, we explore the evolution of cooperation with a different learning process. These results are shown in **Supplementary Figure 3**. We also briefly describe these new robustness checks in the main text.
- 2) Again, to address comments made by all three reviewers, we have extended our discussion section. While our article explores the effect of *incomplete* information, we now also discuss how *asymmetric* information may affect the dynamics. Moreover, we explain the contribution of our study and its implications in more detail.
- 3) Based on comments of reviewer #2, we have revised our Methods section. In the revised version, we explain our computational approach more clearly, and we briefly describe how our algorithms can be extended to capture games with discounting.
- 4) We have also carefully revised our Supplementary Information (SI). Motivated by reviewer #3, we discuss the games for which we observe a benefit of ignorance (**Supplementary Table 1**). Based on comments of reviewer #2, we have revised our description of Lemma 1, and we include a table that summarizes our notation (**Supplementary Table 2**). In addition, we have reduced overlaps with the Methods section, by shortening respective sections in the SI and by moving some of the materials from the Methods to the SI.
- 5) As suggested by reviewer #3, we discuss some additional literature on the value of information in economics and on partial observability in our revised introduction section.

In addition to these major changes, we have corrected several smaller typos and mistakes in some of our figures. We thank the reviewers for their careful observations, and for triggering these changes. We believe the manuscript has improved substantially as a result. Please find a detailed point-by-point response to the reviewer comments below.

Reviewer #1:

This fascinating paper studies the effect of environmental information on the degree of cooperation that evolves in repeated games, in a scenario where there is switching between payoff matrices in a manner that depends on the behavior of the participants in the game. This is a natural extension of previous work on iterated games and environmental feedback, and the authors find counterintuitively that having access to environmental information does not necessarily result in higher levels of cooperation. They systematically classify all the “deterministic stochastic games” (i.e. stochastic games for which the probability of switching between payoff matrices is 1 or 0 depending on the actions of the interacting players in the previous round) as benefitting from the availability of environmental information or not, using a mixture of analytical results (under weak selection and weak mutation), and via simulation (under stronger selection). This work is a valuable contribution to the study of cooperation and has implications that will be of relevance to readers from a wide variety of disciplines. I therefore support publication following revision.

Reply: We appreciate the positive feedback. We would also like to thank the reviewer for the helpful comments below.

Detailed comments

1) The term “deterministic stochastic games” is a bit odd. Indeed the title refers to “stochastic games” but the authors mainly study deterministic switching. I appreciate the use of the term stochastic is consistent with what has come before, but you might want to rethink having it in the title.

Reply and changes: We agree; the term “stochastic games” might seem a bit odd when transitions are deterministic. However, as mentioned by the reviewer, we use the term consistently with the previous literature. Moreover, in addition to cases with deterministic transitions, we also discuss many examples with a stochastic component (**Figure 5, Supplementary Figures 7-12**).

While we have considered several alternative titles, all of them seemed to have certain weaknesses. Overall, we believe the content of our article is most clearly communicated if we stick to the original technical term, by referring to “stochastic games”. However, in the revised main text, we briefly note that stochastic games with deterministic transitions are just a special case of the general framework (lines 89-92).

2) Only deterministic memory-1 strategies are studied. It’s unclear to me whether allowing non-deterministic strategies would disrupt the results i.e. are the results for the timeout game with conditional return critically dependent on the interaction between the deterministic switching and the deterministic strategies? Another perhaps easier way to get at this would be to systematically vary the error rate in the player’s strategies. I appreciate that Fig. 5 looks at an almost-deterministic game and finds qualitatively similar results to Fig. 2, but it’s not clear to me how the system will behave when behavior is not highly deterministic.

Reply: Indeed, in our original manuscript we assumed that all individuals use deterministic memory-1 strategies. This assumption has the advantage that the corresponding strategy set is

finite. For finite strategy sets, the mutation-selection equilibrium can be computed efficiently in the limit of rare mutations. This allows us to compute equilibrium outcomes for many different stochastic games, and for many different parameter combinations. However, we fully agree with the reviewer that it would be interesting to explore to which extent our qualitative results depend on the assumption of deterministic strategies.

Changes: We have addressed this comment in two ways. First, we have run further simulations for the two games depicted in **Figure 2**, allowing individuals to choose among all stochastic memory-1 strategies. We observe a similar ranking as in the baseline case. In the timeout game, full information leads to more cooperation. In the timeout game with conditional return, there can be a benefit of ignorance (see our new **Supplementary Figure 1a,b**).

Second, we have systematically explored the impact of errors when players use pure memory-1 strategies (**Supplementary Figure 2c,d**). Again, we observe the same ranking as in **Figure 2**, provided the error rate is sufficiently small. For larger error rates, the dynamics is more complex.

3) When there is a cost to information, do non-informed strategies outcompete informed strategies i.e. does ignorance evolve?

Reply: This is a great question. In our article, we compare two scenarios. In one scenario, all population members know the precise state of the game ('full information'). In the other scenario, individuals lack any state information ('no information'). As the reviewer suggests, interesting questions arise if we allow for information asymmetry, such that some population members know the state whereas others do not. Such an assumption could have remarkable effects on the strategic interactions. For example, players with knowledge of the present state could adapt their behavior to communicate their knowledge to their interaction partner. Alternatively, such players could try to take advantage of their knowledge. Which of these cases occurs might depend on the exact game being played in each state, and on the game transitions. As the reviewer mentions, the dynamics might additionally depend on whether or not the collection and retrieval of information is costly.

Changes: We believe the issue of information asymmetry is fascinating. At the same time, we feel that a comprehensive treatment of these questions warrants a separate paper that considers the resulting dynamics in detail (the impact of information asymmetry, its evolution, and the impact of information costs). Given that our main text and the SI already contain a substantial amount of material, we thus decided not to tackle the impact of information asymmetry here. However, we have revised the discussion section to explain how information asymmetries could be implemented in our model, and which consequences they might have (lines 355-364).

4) We have found (Stewart, Parsons and Plotkin, PNAS, 2016) that when we compare strategies that have little choice (cooperate or defect) to strategies that have lots of choice (choose how much to cooperate between 0 and full) that there can be a cost to choice, depending on the returns for cooperation. The intuition for this deleterious effect of choice is that increasing choice effectively generates sub-optimal fitness peaks, which the population can get stuck on. It's not

necessary to address our paper specifically, but it seems likely a similar interpretation can be used for the results here and it might be worth noting.

Reply: Indeed, in some of our examples we observe a similar effect. Also in our case, a richer strategy space can sometimes lead to reduced cooperation, albeit for a somewhat different reason (see our **Figure 3c**): In our study, the richer strategy space sometimes allows for the almost neutral invasion of cooperative strategies.

Changes: We now mention the similarity between the two papers in our revised manuscript.

5) It's not at all clear that the indicator function Eq. 4 is represented correctly in Figure 4a. Presumably the x-axis in Fig. 4a is the transition vector q ? Why then are $(1,1,1;1,1,1)$ and $(0,0,0;0,0,0)$ at the right-hand end of the x-axis not classified as neutral? Clearly information is not beneficial in either of these cases and indeed the indicator function for $(1,1,1;1,1,1)$ should be $X=(1+0)-(1+0)=0$. I suspect some axis mislabelling has occurred.

Reply and changes: Thank you for making us aware of these issues, we have fixed them.

6) The panel labelling in Figure 2 is also messed up

Reply and changes: The reviewer is correct, thank you for making us aware of this.

Alex Stewart

Reviewer #2:

The manuscript studies the effect of environment information on emergence of cooperation using the stochastic games framework. In particular, the framework considers two players in a stochastic game with two actions (Cooperate, Defect) and two states (profitable, less profitable for cooperation). The evolutionary behavior follows a smoothed best response with some noise (error) in choosing among possible memory-1 policies. In the full information case, players observe the current state and performance of other strategies in the last step. In the no information case, players only observe the performance of other strategies in the last step, but they cannot condition their current policy on the state of the environment. The paper provides a systematic analysis of the arising cooperation levels in full and no environment information settings considering the benefit of cooperation in the profitable state, selection strength, and transition probabilities. The results point to a preference for environment information (full information) in emergence of cooperation. Specific scenarios in which no environmental information leads to higher cooperation levels are rare, and tend to have limited improvement in cooperation levels over the full information settings.

The paper considers the emergence of cooperation in populations by incorporating relatively unexplored features in prior works: environment feedback, and information availability. The

framework design is simple, yet expectedly exhibit rich set of dynamics. Most of the results are obtained by an exhaustive simulation of the entire set of outcomes. A few of the results rely on analytical derivations. Other analytical results simplify the search space, easing the computational burden of exhaustive simulation. The paper is well-written with a clear and relevant motive. The take-away results are often mixed, in the sense that there is no clear winner between the two options (full or no information). Overall, the paper takes a systematic approach to understanding the emergence of cooperation in relatively unexplored settings.

Reply: Thank you for this summary of our work, and for the very constructive comments below.

1) My main comment is that there is no guarantee provided for the convergence of the learning dynamics in the proposed setting. Without such a guarantee, the outcomes from the simulations and interpretations could depend on the behavior dynamics. Indeed from Fig. 3, it seems the population can shift between different policies. I.e., there exists a set of policies that are not eliminated by the selection procedure. In this case, the relation between the emergence of cooperation and the amount of time spent in the preferred state can become tenuous. I realize the analysis focuses on the relative cooperation rates for populations with full and no information, but a note on the convergence properties of the learning dynamics and its relation to surviving policies is necessary.

Reply: Thank you for making us aware of this issue. To describe the learning dynamics in our populations, we use a stochastic process. For finite selection strength (i.e., when there is always a positive probability to adopt an inferior strategy), this process does not have any absorbing states. This means that the dynamics never converges to a particular state. However, because the stochastic process is ergodic, the time averages converge (i.e., the probability to be in any given state over the course of time). By the theorem of Perron-Frobenius, these time averages can be derived by computing the invariant distribution of the process. This is what we do to compute the long-run abundance of each strategy, and the resulting average cooperation rates.

Changes: The reviewer's comment made us realize that we did not specify our computational methods in sufficient detail. We have therefore carefully revised our main text and the Methods section, to clarify in which sense our process converges.

2) There are other learning dynamics that can be considered. Should we expect the results to change as the learning dynamics change? In other words, are the results robust to changes in the learning dynamics?

Reply: This is a fair point. Throughout our article, we assumed that successful strategies propagate in a population by imitation. To model imitation, we use a pairwise comparison process. While the pairwise comparison process has become a standard model in evolutionary game theory, it is natural to ask to which extent our results depend on this specific process.

To explore this issue in more detail, we revisited the two games considered in **Figure 2**. Instead of an imitation process, we explored a learning process that is based on introspection. This introspection dynamics has been introduced recently to explore strategic-decision making in asymmetric games (Hauser et al, Nature 2019). However, it can be equally applied to symmetric games (Couto et al, New J. Phys. 2022). Introspection dynamics considers a single pair of players. At each time step of the learning process, one of the players is randomly chosen and given an opportunity to revise its strategy. To this end, the player compares its current payoff with the hypothetical payoff the player could have obtained by choosing some alternative strategy. The higher the payoff of the alternative, the more likely the player switches.

The two processes differ in important aspects. For example, the pairwise comparison process describes strategy updating in evolving populations. In contrast, introspection dynamics describes learning among two interacting players. As a result of these differences, the two processes predict different absolute cooperation rates for the two games depicted in **Figure 2**. Importantly, however, our qualitative findings remain unchanged. For the first game ('timeout game'), we observe a benefit of information for all considered parameter values. In contrast, for the second game ('timeout with conditional return'), there can be a benefit of ignorance.

Changes: We show the new simulation results for introspection dynamics in **Supplementary Figure 3**. In addition, we briefly mention this figure in the main text.

3) In the SI, there is a mention on how we can rely on a subset of the policies available at the full information setting to study the no information setting. This is intuitive, since some of the policies in the full information can choose to ignore state information, and thus mimic the no state information setting. Perhaps this property can be used to make statements about asymmetric information where one set of agents is given the state information and the other set is not.

Reply: We agree, this is a very interesting question (which was also brought up by reviewer #1). In our paper, we compare two basic scenarios. Either every population member knows the precise state of the environment, or no one. It would indeed be fascinating to explore what happens in between these two boundary cases. For example, once one player has information that its opponent has not, this player may try to take advantage of the information asymmetry. Whether or not this player can do so may depend on the exact games being played in each state, and on the state transitions between them.

Change: While information asymmetries pose interesting new problems, we feel that a satisfactory treatment of information asymmetries would provide material for an entirely different paper. Given that our main text and our SI already contains a substantial amount of material, we decided to keep our focus on symmetric games. However, we provide an informal discussion of the possible role of information asymmetries in our revised discussion section (lines 355-364).

4) A major modeling assumption is that there is no discounting of future payoffs. The discounting could be a significant factor in determining the relative benefit of information. This assumption and its potential implications should be clearly specified and discussed in text.

Reply and Changes: We agree that such an assumption should be stated more clearly, and we do this now both in the main text and in the methods section. In addition, for the games considered in **Figure 2** of the main text, we study the effect of varying a discount factor δ in a new set of simulations. As one may expect, we find that for δ sufficiently close to one, the qualitative results are similar to the case in the theoretical limit $\delta \rightarrow 1$. The outcome of these simulations is shown in **Supplementary Figure 1e,f**.

5) Proposition 3's proof is by Mathematica which is not a valid argument/proof.

Reply and changes: Point taken; we re-wrote this part.

6) Many times the reader is referred to the SI within the text. However, it is not clear which part of SI is related. The in-text referrals to the SI should be more specific. It should specify which equation/result/figure the reader should see for ease of following the flow.

Reply and changes: This is a very helpful suggestion; we have adapted the main text accordingly.

Some other minor comments follow:

7) There is a significant overlap between SI and methods. In particular, SI Sections 1.2 - 1.4 and parts of Section 2 appear in methods verbatim. There is a lot of redundancy.

Reply and changes: We agree, there were quite a few redundancies. In the meantime, we have revised both the Methods section and the SI to minimize this overlap.

8) Lemma 1's statement is not easy to follow given that it requires flipping through several pages before to be reminded of the definitions. Either remind the reader of the definitions within the lemma's statement or in text right before the lemma's statement.

Reply and changes: Thank you for making us aware of this issue. We agree, because of the organization of our SI, readers might find it difficult to follow this section. We have rephrased the statement of Lemma 1, and we have added a table of notation (**Supplementary Table 2**).

9) There is a reference to Fig. 4d which does not exist.

Reply and changes: This is correct, this was an error on our part. We have fixed this now.

10) Please make sure all acronyms, e.g., ALLD, TFT, are defined the first time they are used.

Reply and changes: Thank you, fixed.

Reviewer #3:

****Summary:**** The authors investigate the influence of state information on the evolution of cooperation in stochastic games. Specifically, given the environmental transition function, they investigate whether memory-1 strategies that condition on the environmental state (i.e., taking state information into account) are better, neutral, or worse than a state-unconditional memory-1 strategy. Their evolutionary model consists of the following parameters: the selection strength β , the mutation rate μ , and the strategy-error rate ϵ . The manuscript provides an extensive analysis (combining analytical with simulation results) of deterministic state transitions in coupled two-state, two-agent, two-action Prisoner's dilemmas. Extensions to stochastic transition functions and three-agents public good games are also considered. Overall, the authors find the influence of state information on the evolution of cooperation in stochastic games to be non-trivial, i.e., it can be beneficial, neutral, and harmful, with non-trivial dependence on the model's parameters.

****Overall assessment:**** Overall, I enjoyed the paper, which makes a relevant contribution. It is well written and provides an extensive analysis that robustly underpins the main finding of the work. Yet, I believe some improvement can be made. These concern a clarification and contextualization of the model and its main findings (Comments 1-5) and some clarifications and suggestions regarding specific aspects (Comments 6-8).

Reply: Thank you for the encouraging feedback, and for the constructive feedback below.

****Specific comments****

1) In its current form, the main take-home message is that the effect of state information on the evolution of cooperation in stochastic games is non-trivial. While this is an important finding, my impression is that more can be said from all the analyses. The work would benefit from distilling its non-trivial analyses and results into relatable conclusions. For example, I would be excited to see a summary of the conditions when ignorance promotes the evolution of cooperation in stochastic games. And when it does, when is it even better than conditional strategies? Currently, the authors only give one example (Fig. 2e-h). But from Figs. S1-S3, I believe, games 51-55 are also possible candidates, among others. What can be said about the similarities between their transition function? How do the evolutionary parameters β , μ , ϵ affect the outcomes?

Reply: We agree, our framework can lead to very non-trivial dynamics, and we consider this to be one of the major insights of our study. In addition, our study provides an exact criterion for when we observe a benefit of ignorance in the limit of weak selection. Further computational analyses for stronger selection strengths suggest that cases with a benefit of ignorance should be comparably rare overall.

Of course, it would be desirable to have a simple rule of thumb (for all selection strengths) that predicts a game's benefit of information. Unfortunately, our analysis suggests that there might be

no such rule. Instead, we often observe an intricate interplay between the transition function of the game, as well as all the game's parameters. To illustrate this observation, the table below summarizes all games that promote ignorance for sufficiently large selection strength and benefit in the first state:

Game ID	Transition function	Maximum benefit of ignorance
38	(100; 110)	0.0904
44	(101; 100)	0.0001
46	(101; 110)	0.1288
52	(110; 100)	0.0014
53	(110; 101)	0.0188
55	(110; 111)	0.0191

As this summary suggests, a benefit of ignorance is rare for deterministic transition functions. Moreover, the magnitude of the effect tends to be small. The major exceptions are the games 38 (100; 110) considered in **Figure 2** and the game 46 (101; 110). The transition function of game 46 differs from the game 38 only in one entry (in game 46, players stay in the beneficial state 1 even if they both defected).

For all the games in this table, mutual cooperation leads the two players to the more profitable state (irrespective of the current state). However, beyond this pattern, the transition functions have very little in common. Moreover, among all 16 cases where mutual cooperation leads to the more profitable state, the above six cases are only a subset.

Changes: As mentioned above, we are unable to provide a simple rule that predicts the benefit of information of any given game. However, we feel that a more detailed discussion of this topic is useful. We provide this discussion now in the revised main text, and in more detail in the new **SI Section 4.3**, which also contains the above table (**Supplementary Table 1**). In addition, we discuss the impact of the parameters ε and μ in our new **Supplementary Figure 1**. As shown there, changes in the error rate and the mutation rate have little impact on the results as long as these changes are sufficiently small.

2) What are the consequences of this work? Despite the conceptual nature of this model, it would be interesting to the readers to hear about possible consequences the authors envision from their study. Be it a theoretical advancement on the evolution of cooperation in stochastic games, a possible explanation why ignorance (about the state information) exists, or eventually looking for cooperation-promoting mechanisms that utilize the form of ignorance studied in this work.

Reply: Thank you for raising this question. We would argue that our work contributes to the literature in at least two ways. For one, we offer a useful theoretical framework that allows modelers to explore the impact of information on the evolution of cooperation. This framework can be extended to various directions. For example, as also suggested by the other reviewers, researchers might use this framework to further explore the effect of asymmetric information in stochastic games (such that some but not all players know the state of the environment).

In addition, our study highlights the importance of information in games in changing environments. For most games that we studied information is beneficial, and the value of information can be quite substantial. We believe this is quite an important message that can serve as the baseline for future studies.

Changes: We have revised the discussion section to put more emphasis on the implications of our study, and possible follow-up research.

3) Regarding the appropriate referencing to previous literature, the manuscript would benefit from a discussion on how the presented approaches relate to

1) the value of information in economics, e.g.,

- Levine, P., & Ponsard, J. P. (1977). The values of information in some nonzero-sum games. *International Journal of Game Theory*, 6(4), 221-229.

- Bagh, A., & Kusunose, Y. (2020). On the economic value of signals. *The BE Journal of Theoretical Economics*, 20(1).

2) the framework of partial observability, e.g.,

- Hansen, E. A., Bernstein, D. S., & Zilberstein, S. (2004, July). Dynamic programming for partially observable stochastic games. In *AAAI* (Vol. 4, pp. 709-715).

- Barfuss, W., & Mann, R. P. (2022). Modeling the effects of environmental and perceptual uncertainty using deterministic reinforcement learning dynamics with partial observability. *Physical Review E*, 105(3), 034409.

Reply and changes: Thank you for the suggested literature. While there is indeed a substantial literature in economics and computer science addressing the value of information, the evolutionary literature to date has neglected these aspects. The proposed literature is close in flavor to our setup, yet there are significant differences. We included the corresponding citations in our manuscript in our introduction.

4) The authors study the evolution of cooperation of memory-1 strategies either with or without additional conditioning on the environmental states. Doing so enables two possible reasons for the evolution of cooperation, conditioning on the environment and conditioning on the previous actions. To better contextualize the results, it would be helpful to compare them to two baselines. First, conditional memory-zero strategies, which only condition on the environmental state, might evolve to cooperate. Second, how do memory-one strategies evolve in the static game without environmental transitions?

Reply: Thanks for the feedback! We agree, when interpreting our results, it is useful to have a baseline. Therefore, we ran the additional simulations suggested by the reviewer. As one may expect, we found that memory-zero strategies do not lead to cooperation in the stochastic games. This observation suggests that for the games and parameter values we consider, reciprocity is necessary for cooperation to evolve. Regarding the evolution of memory-one strategies in static games without environmental transitions, we recover the respective results of Hilbe et al. (*Nature*

2018). In particular, we show that stochastic games can lead to cooperation even if cooperation does not evolve in either of the stage games.

Changes: We present the new simulation results in **Supplementary Figure 2**. In addition, we briefly summarize the respective findings in the main text.

5) The authors provide an extensive mathematical analysis. However, sometimes the manuscript reads as if some modeling assumptions have only been taken to be able to execute some mathematical techniques. For example, the manuscript investigates the weak selection limit multiple times, as often done in evolutionary analysis. But I would welcome a justification or reasoning for why this is interesting, other than analytical tractability. For example, the yellow boxes in Fig. S1 clearly show that the results of the weak selection limit do not necessarily carry over to non-zero selection strengths. Another modeling assumption that is introduced rather ad-hoc is the error rate ϵ . I believe the authors have done so to avoid non-ergodic Markov chains when calculating the payoffs. While this is a valid reason, what modeling reasons apply, such as the agents' imperfection in carrying out a deterministic strategy or the need for exploration as frequently considered as the exploration-exploitation trade-off in reinforcement learning?

Reply: We agree that the model rests on a set of assumptions and partly, these assumptions are made for mathematical convenience. However, we note that both of the assumptions mentioned by the reviewer have a long tradition in the evolutionary game theory literature.

For example, the limit of weak selection is commonly used to obtain analytical expressions for the abundance of strategies (e.g. Traulsen and Hauert 2009). These expressions can then be further analyzed to explore the impact of parameter changes on the evolutionary dynamics. We derive the respective results for a selection strength of $\beta = 0$. However, by continuity it follows that qualitatively similar results hold when β is positive but sufficiently small (the only exception occurs in those cases in which information is exactly neutral for $\beta = 0$). We complement these analytical results for small selection strengths with numerical results for arbitrary β .

A similar argument applies to execution errors. One reason to allow for these errors is indeed a technical one. Execution errors ensure that the players' average payoffs are well-defined, irrespective of the players' initial moves (e.g. Sigmund's book 'Calculus of selfishness', 2010). In addition, however, execution errors also make the model more realistic. For example, people may intend to cooperate, yet they may fail to do so because of a mistake. Again, such models with execution errors have a long tradition in evolutionary game theory (e.g., Boyd, JTB 1989).

Changes: In our revised manuscript, we provide a better motivation for the assumptions of weak selection and of positive error rates. In addition, we now illustrate the impact of different error rates in **Supplementary Figure 1c,d**.

6) It did not become clear to me why the authors chose to study almost-deterministic transition functions. What is special or interesting about this type?

Reply: We started our analysis with the very simple deterministic transitions to gain some intuition for the value of information. Yet, while mathematically convenient, such transitions lack natural stochasticity of the environmental transitions. From this class of transitions, we can expand our analysis to consider vectors with one entry being a continuous variable q in $[0,1]$. We referred to this class almost-deterministic transitions, but perhaps a better name would be single-stochastic transitions. This class includes 192 families of games; for every value of q we obtain a different game. Single-stochastic transitions allow us to account for some degree of environmental stochasticity while keeping the model tractable.

We provide additional results for fully stochastic transition vectors in **Supplementary Fig. 12**. There we assume that transition probabilities are taken from the set $\{0.0, 0.2, 0.4, 0.6, 0.8, 1.0\}$. Even with this simplifying assumption, there are now more than 46,000 games to consider. While we believe this analysis gives a great overview, it is impossible to provide a complete and meaningful systematic analysis of all such transitions, given the sheer number of cases.

Changes: Throughout the manuscript, we replaced the notion of “almost-deterministic games” with “single-stochastic transitions”. In addition, we motivate more carefully why we consider single-stochastic transitions, in addition to fully deterministic and fully stochastic transitions.

7) Something must have gone wrong with the labeling of Fig. 4. I see only three subfigures, a,b, and c, while the caption refers to four: a,b,c, and d. I believe Eq. 4 was part of the Figure at some point. Also, the caption has a different symbol for the value of information than the main text. Furthermore, I'm unable to replicate the relationship between the transition functions (as the labels on the x-axis) and the value of X (Eq. 4). For example, let's focus on $X=2$, which can only be achieved if the transition function $(q1_CC, q1_CD, q1_DD, q2_CC, q2_CD, q2_DD)$ as given in Eq. 1 is (1_00_1) with $_$ either 0 or 1. But the four transition functions in Fig. 4a associated with $X=2$ are $(111101, 111110, 111111, 000000)$.

Finally, in the text, Eq. 4 is introduced to apply to the 32 non-neutral cases. However, Fig. 4a shows all 64 cases, with the neutral transitions displayed as $X=0$. This suggests that Eq. 4 actually applies to all 64 cases.

Reply and changes: Thank you for making us aware of these issues. We have fixed them. Regarding the second point, the significance of the proxy function X is as follows. For all cases in which information is non-neutral, we can show that either $X>0$ or $X<0$. Moreover, in most cases in which information is neutral, one can show that $X=0$. The only exception occurs when the stochastic game has an absorbing state. These games are always neutral, irrespective of their X -value. Here, players know that they eventually end up in the absorbing state. Therefore, the two scenarios of ‘full information’ and ‘no information’ become equivalent.

8) For the large table figures in the supporting information, it would be helpful if they were ordered according to the classes introduced in the main text: neutral (absorbing states, state-independent transitions, symmetries) and non-neutral ($X>0$, $X<0$).

Reply: We appreciate the suggestion to reorganize our figure panels. Unfortunately, we believe that it would become difficult for the readers to keep track of the games and their dependence on parameter changes if we order them by classes. For instance, the deterministic game 42 and the single-stochastic game 99 will belong to different classes, depending on the parameter we vary (selection strength, benefit in state 1, or error rate). There are many more examples of games that exhibit a similar change in their qualitative behavior. In light of these problems, we felt that our presentation is most transparent if we order the different transition functions lexicographically, according to their transition function.

Reviewers' Comments:

Reviewer #1:

Remarks to the Author:

The authors have made substantial changes to the manuscript and either addressed my questions, or else provided a satisfactory response in their replies. Therefore I am happy to support publication.

Reviewer #2:

Remarks to the Author:

Thank you for addressing my comments/questions. I find the model and results clearly explained and novel.

Reviewer #3:

Remarks to the Author:

The authors responded satisfactorily to all my points, and I agree that they could improve the manuscript through the revision round. My only issue is that it would have been great to view a copy of the updated manuscript with track changes.